# The Mediating Role of Work Engagement in the Relationship between Executive Functioning Deficits and Employee Well-Being

**DOI:** 10.3390/ijerph192013386

**Published:** 2022-10-17

**Authors:** Chee-Seng Tan, Hira Nasir, Kai-Shuen Pheh, Chin Wen Cong, Kok-Wai Tay, Jia-Qi Cheong

**Affiliations:** 1Department of Psychology and Counselling, Faculty of Arts and Social Sciences, Universiti Tunku Abdul Rahman, Kampar 31900, Perak, Malaysia; 2Faculty of Business, Economics and Accountancy, Universiti Malaysia Sabah, Jalan UMS, Kota Kinabalu 88400, Sabah, Malaysia; 3Centre for Economic Development and Policy, Universiti Malaysia Sabah, Jalan UMS, Kota Kinabalu 88400, Sabah, Malaysia

**Keywords:** creativity, employee, engagement, executive function, job demands-resources model, Malaysia, workers, well-being

## Abstract

Executive functioning and its related components have been found to promote well-being. However, there is a limited understanding of the underlying mechanism. Drawing from the job demands–resources and PERMA models, the present study examined the hypothetical mediating role of work engagement in the relationship between executive functioning deficit and well-being among 314 working adults in Malaysia. Participants answered a survey consisting of the Executive Skills Questionnaire-Revised (ESQ-R; a new measure of executive functioning deficits for working adults), Utrecht Work Engagement Scale, Employee Well-Being Scale, and Self-Rated Creativity Scale. Pearson correlation analysis showed that the ESQ-R score was negatively associated with all other target variables, while the latter was positively related to each other. Moreover, supporting the hypotheses, the results of mediation analysis using PROCESS macro found that work engagement mediated the negative relationship between executive functioning deficits and well-being after statistically controlling for the creativity score. The findings not only replicate the beneficial role of executive functioning in employees’ well-being but also shed light on the underlying process of the relationship. Implications and directions for future studies are discussed.

## 1. Introduction

Executive functioning (EF) is a set of mental abilities that “optimize the efficiency and effectiveness of behavior, allowing behaviors that are more goal-oriented, autonomous, and conceptually driven” [1]. Baggetta and Alexander systematically reviewed the literature and indicated that EF involves several higher cognitive processes [2], including managing automatic reactions (i.e., inhibitory control; [3]), retaining and manipulating information in the mind (i.e., working memory; [4]), switching between multiple tasks and mental processes to generate suitable responses (i.e., shifting; [5]; cognitive flexibility; [6]), focusing and sustaining attention on a particular task for a period (i.e., attentional control), managing and responding adaptively to emotional experiences [7], and other metacognitive skills. Diamond has proposed an EF model that conceptualizes inhibitory control, cognitive flexibility, and working memory as the core components of EF, as well as the foundations of higher-level EFs, such as planning and fluid intelligence [8]. Empirical studies show that EF is beneficial to emotional regulation [9,10], creativity [11], acute stress responses [12], self-regulated learning [13], weight loss and lifestyle modification [14], academic engagement [15], and subjective well-being [16].

While EF has been extensively studied in the clinical populations and its psychopathological implications (e.g., [17,18,19]), it receives relatively little attention in the workplace setting. This study establishes empirical evidence of the relationships between executive functions, work engagement, and employee well-being. We hypothesize there is a significant relationship among the abovementioned variables.

### 1.1. The Relationship between Executive Functions and Employee Well-Being

The empirical findings of a positive relationship between EF and emotion regulation (e.g., [9,10]) suggests that EF are beneficial to well-being. Indeed, there is a growing body of research that supports a positive relationship between EF and well-being. For instance, EF-related constructs such as strategic planning [1], self-control [20], and the ability to update positive information in working memory [21] have also been associated with well-being and/or its domain (e.g., life satisfaction).

Toh and colleagues analyzed responses of 3267 individuals of age between 32 years old to 84 years old, found that EF is positively correlated with life satisfaction, and the relationship is mediated by a sense of control [16]. The findings are consistent with the mechanism outlined by Luerssen and Ayduk that the ability to inhibit automatic impulses (i.e., self-control skills) such as overriding self-interested impulses, inhibiting impulses to junk food consumption and managing distraction is related to domains of well-being (e.g., healthy relationships, good health) [22].

The beneficial effect of EF is also expected in the workplace. As distractions and temptations hinder employees’ attention and delay the completion of tasks on hand, it is reasonable to believe that employees with good inhibitory EF to manage distractions can complete tasks successfully and hence, tend to have greater well-being. In a study involving 482 full-time employees in China, self-control was positively correlated with well-being in terms of job satisfaction and life satisfaction [23]. Moreover, Dou and colleagues found that self-control has an indirect relationship with life satisfaction via job satisfaction [23].

The positive relationship between EF and well-being can be explained by the job demands–resources (JD-R) model [24,25]. Job demands are the physical and/or psychological obstacles in achieving work goals and have been constantly found to have negative impacts on employees (e.g., [26,27,28,29]). Alternatively, job resources refer to physical, psychological, social, or organizational aspects that are functional in achieving work goals and hence are advantageous to employees [24,28]. The hypothetical negative effect of job demands and the positive effect of job resources have been evident in empirical studies. For instance, analysis of the self-reported job demands and resources, work engagement, and well-being of 353 cruise ship officers and staff, Radic and colleagues found that job demands are negatively associated, while job resources are positively associated with well-being [30].

The JD-R model has been expanded to include personal resources, the psychological characteristics that are helpful for employees to boost job performance and handle the encountered challenges successfully [29]. Xanthopoulou and colleagues analyzed data collected from 714 Dutch employees and found that job resources are positively associated with work engagement and negatively associated with exhaustion [31]. Moreover, personal resources (indexed by self-efficacy, organizational-based self-esteem, and optimism) were found to play a mediating role in both relationships. The findings indicate that, like job resources, personal resources are beneficial to employees. In the same vein, Bakker and de Vries propose a new model to explain the development and decrease in job strain and burnout by incorporating the JD-R model and the findings of self-regulation [32]. They postulate that when experiencing job strain, employees tend to use more maladaptive self-regulation strategies (e.g., self-undermining) and less adaptive self-regulation strategies (e.g., job crafting). The latter is assumed to enhance job and personal resources, which in turn, can decrease job strain. In other words, job and personal resources and self-regulation are helpful to employees’ well-being by reducing job strain and burnout. Meanwhile, as EF is a set of self-regulation abilities, it is reasonable to hypothesize a positive relationship between EF and well-being based on the JD-R model. 

### 1.2. The Relationship between Executive Functioning and Work Engagement

The JD-R model also offers support to the hypothetical (positive) relationship between EF and work engagement, an enjoyable and positive mental state at work in terms of vitality, enthusiasm, and concentration [33]. Work engagement has been found to have a positive relationship with job-related constructs (e.g., job commitment, performance) and personal well-being (e.g., life satisfaction, health) in a meta-analysis [34]. As reviewed above, job resources, as well as personal resources, are helpful for workers to cope with challenges and engage in the assigned tasks. Indeed, studies have consistently found a positive relationship between job resources and work engagement [30,31,35]. 

Similarly, employees who have high EF are likely to manage their resources (e.g., attention) to focus on tasks and hence, can engage in work. Although few studies have directly examined the relationship between EF and work engagement, various EF-related dimensions (e.g., self-management, time management, and emotional regulation) have had a positive relationship with work engagement [36,37,38]. For instance, Greenier and colleagues examined 108 British and 255 Iranian English language teachers and found that those who are effective in regulating feelings reported higher involvement in their teaching emotionally, cognitively, and psychologically [39]. 

### 1.3. The Relationship between Work Engagement and Employee Well-Being

Literature has suggested that engagement is beneficial for individuals’ well-being. According to the PERMA model [40], engagement has been proposed as one of the five elements that contribute to people’s flourishing, a construct that has been used interchangeably with well-being. 

The positive association of engagement and well-being has also been documented in the work setting. Work engagement is essential for employees to attain self-fulfillment. Workers with high work engagement tend to have higher levels of autonomy and commitment at work [41]. With those benefits, engaged workers are keen to dedicate their energy and devotion toward their jobs, besides exhibiting great vitality, enthusiasm, and concentration. The self-improvement and self-realization resulting from positive working characteristics are advantageous to employee well-being.

Empirical studies have consistently evidenced a positive relationship between work engagement and employee well-being, including its relevant domains (e.g., life satisfaction) in different cultural contexts (e.g., [42,43,44,45,46]). In a two-year longitudinal study involving 1196 employees from an industrial machinery company in Japan, Shimazu and colleagues found that the three subdomains of work engagement (i.e., vigor, dedication, and absorption) are positively correlated with improvement in psychological well-being, as indexed by job satisfaction and family satisfaction (*r* = 0.09–0.16) [47]. Furthermore, structural equation modeling (SEM) results also showed a positive relationship between the latent construct of work engagement and positive changes in life satisfaction [47]. Similar results are also observed among 208 pairs of working parents with preschool children in Tokyo [48]. Correlation analysis showed that work engagement was positively associated with happiness for both parents. However, further analysis using SEM found a direct pathway from work engagement to happiness among fathers but not mothers [48]. The authors attribute the results to gender role differences. Fathers in Japan devote more time to work-related activities (i.e., less time to housework and child-rearing) than their spouse, and hence, work engagement not only has an indirect relationship with happiness via work-to-family facilitation, but also a direct relationship. 

Work engagement has also had a longitudinal, beneficial effect on employee well-being. Yang and colleagues collected data from full-time employees working in China using a 3-wave survey across two months to examine the direct impact of career adaptability on employee well-being and the indirect effect via work engagement [49]. It was found that work engagement measured at the second wave had a positive effect on employee well-being measured at the third wave with a 1-month interval [49]. 

### 1.4. The Present Study

As reviewed above, both theoretical frameworks (e.g., JD-R model and PERMA model) and empirical evidence suggest that the three key constructs of this study—EF, work engagement, and well-being—are related to each other. Moreover, the literature also suggests that EF has an indirect relationship with well-being through work engagement. However, to our best knowledge, this underlying mechanism has not been empirically tested. To address this theoretical gap, we integrated the data of Nasir and colleagues’ study [50] and newly collected data to examine the hypothetical mediating role of work engagement in the relationship between EF and employee well-being in a sample of working adults in Malaysia. Self-rated creativity was included as a covariate variable in the analysis because creativity is related to EF [11], work engagement [51], and well-being [52], respectively. The findings are expected to shed light on the benefits of EF on well-being in the working contexts and provide insights into the underlying mechanism of this relationship. 

## 2. Method

### 2.1. Research Design and Participants

A total of 340 participants, of which 324 participants were obtained from Nasir and colleagues’ study [50], were recruited using the convenience sampling method, where they completed an online survey. After removing 26 cases with empty responses on at least one of the four scales used in this study, the final sample for analysis consisted of 314 working adults (298 were from Nasir and colleagues’ study [50]) who were working in Malaysia (54.8% females, one missing value). The average age of the samples was 39.60 (*SD* = 10.10), ranging from 23 to 80 (five participants did not report their age). Besides, the sample reported an average of 9.20 years of working experience (*SD* = 7.50), ranging from 1 to 40. Most of them (86.30%) were Malaysians, while the rest (13.70%) were foreigners. Among them, 35.40% were Chinese, 35.0% were Malays, 15.90% were Indians, and 13.70% were from other ethnicities. In terms of occupation (except five participants who did not reveal their occupation), 66.6% of them worked in the academic field (e.g., lecturers and teachers), and the rest worked in non-academic fields such as business and administration (e.g., executive), management (e.g., director), healthcare (e.g., therapist), and legal (e.g., lawyer). 

Before answering the online survey, participants were required to give their consent after reading the general instruction about the study. The university’s Scientific and Ethical Review Committee approved this study.

### 2.2. Measurements

#### 2.2.1. Executive Skills Questionnaire-Revised (ESQ-R)

The ESQ-R [15] was used to assess EF deficits using a 4-point (frequency-based) Likert-type scale, ranging from 0 (never or rarely) to 3 (very often). The ESQ-R consists of 25 items that are grouped into five subscales: plan management, time management, material organization, emotional regulation, and behavioral regulation. The plan management subscale has 11 items that reflect how well an individual is in completing a job by planning (e.g., “I have trouble with tasks where I have to come up with my own ideas”). The time management subscale has four items related to several facets of time management ability, such as estimating time, allocating time, and working within the limitations of time (e.g., “I have trouble estimating how long it will take to complete a task”). The materials organization subscale has three items that measure how well an individual is in developing and keeping systems to stay conscious of materials or information (e.g., “I lose things”). The emotional regulation subscale consists of three items that evaluate how well an individual is in managing feelings for task accomplishment, goal achievement, and behavioral control (e.g., “I get upset when things don’t go as planned”). The behavior regulation subscale with four items assesses an individual’s self-control and thinking of the effects of behavior before acting (e.g., “I say things without thinking”). The mean score of each subscale reflects the executive deficiency in a particular area, while the total score of the five subscales indicates the general executive dysfunction of an individual. Hence, a higher total score signifies a poorer overall EF. To ease understanding, the ESQ-R total score is hereinafter referred to EF deficits.

Nasir and colleagues found that four items (items 4, 6, 19, and 24) of the ESQ-R are not commonly used in the Malaysian context and rephrased those items to ease understanding [50]. For example, item 4 (“I have a short fuse”) was modified into “I tend to get angry easily”. Nasir and colleagues demonstrated that the ESQ-R with the four modified items has a good model fit, internal consistency, and validity (i.e., convergent, discriminant, and concurrent validities) [50].

#### 2.2.2. 9-Item Utrecht Work Engagement Scale (UWES-9)

The UWES-9 [25] with nine items was employed to measure participants’ work engagement using a 7-point Likert-type scale, ranging from 0 (never) to 6 (always/everyday). Sample items for the three subscales of the UWES-9 were “At my work, I feel bursting with energy” (vigor), “I am enthusiastic about my job” (dedication), and “I am immersed in my work” (absorption). A higher total score indicates a higher engagement in work.

#### 2.2.3. Employee Well-Being Scale (EWB)

The EWB [53] was used to assess employee well-being, in which the participants were required to answer 18 items on a 7-point Likert-type scale, ranging from 1 (strongly disagree) to 7 (strongly agree). With three subscales, some sample items of the EWB were “My life is very fun” (life well-being), “Work is a meaningful experience for me” (workplace well-being), and “I handle daily affairs well” (psychological well-being). A higher mean score implies higher levels of employee well-being.

#### 2.2.4. Self-Rated Creativity Scale (SRCS)

Participants responded to the 12-item SRCS [54] on a 5-point Likert-type scale, ranging from 1 (strongly disagree) to 5 (strongly agree), to indicate their perceived creativity. A sample item was “I am a good source of creative ideas”. A higher mean score suggests higher self-perceived creativity.

### 2.3. Analytical Plan

The mediation model of work engagement in the relationship between EF deficits and employee well-being with self-rated creativity as a control variable was analyzed using SPSS PROCESS macro programmed by Hayes [55]. In the analysis, we specified EF deficits as the predictor, employee well-being as the outcome variable, work engagement as a mediator, and self-rated creativity as the covariate under Model 4. A 10,000 bootstrapping was used to establish a 95% bias-corrected confidence interval (CI) to confirm the indirect effect, in which the effect is significant if the CI excludes zero.

## 3. Results

An examination of the dataset (*n* = 314) found three cases with missing values in UWES (item 7), EWB (items 6 and 17), respectively. Since the missing value was not at random (i.e., Little’s MCAR test was significant), the expectation maximization (EM) was not used. Instead, multiple imputations were conducted on the dataset. Thus, all further analysis was based on the imputed data (*n* = 314, imputed). The results obtained from the imputed data were similar to the results obtained from the data without the three cases with a missing value (*n* = 311).

Table 1 shows the descriptive statistics, Cronbach’s alpha coefficients, and intercorrelations for the study variables. All variables were normally distributed as indicated by the absolute values of skewness and kurtosis, which were below one and two, respectively [56]. Reliability analysis revealed that all measures had high internal consistency (Cronbach’s alpha coefficients ranged from 0.91 to 0.95). Pearson correlation analysis demonstrated statistically significant intercorrelations among EF deficits, work engagement, employee well-being, and self-rated creativity, which served as initial support to the hypothesis. Note that the negative relationships among EF deficits and the other variables indicate that individuals who reported a lower EF deficit tend to have a higher level of work engagement, well-being, and creativity.

The mediation analysis showed that self-rated creativity was positively associated with work engagement (unstandardized coefficient [*B*] = 0.63, *t* = 17.36, *p* < 0.001) and employee well-being (*B* = 0.28, *t* = 9.66, *p* < 0.001 in the direct effect model; *B* = 0.58, *t* = 17.99, *p* < 0.001 in the total effect model).

After controlling for self-rated creativity, the total effect of EF deficits on employee well-being was significant, *B* = −0.03, *t* = −14.99, *p* < 0.001. However, EF deficits had a negative relationship with work engagement, *B* = −0.03, *t* = −12.30, *p* < 0.001. The latter was found to have a positive relationship with employee well-being, *B* = 0.47, *t* = 26.86, *p* < 0.001. After controlling for the effect of work engagement, the direct effect (of EF deficits) was also significant, *B* = −0.02, *t* = −9.58, *p* < 0.001. Besides, the indirect effect of EF deficits on employee well-being through work engagement was significant, *B* = −0.01, *SE* = 0.001, CI [−0.014, −0.010]. The results indicated that EF deficit was indirectly and negatively associated with employee well-being through work engagement. Figure 1 shows the mediation model with the standardized coefficients.

As the present study relied on self-report, Harman’s single factor test was conducted to investigate the impact of common method bias on the results. Items of the four measurements were submitted to exploratory factor analysis. By fixing the extraction to one and using principal axis factoring estimation without rotation, the model explained 29.227% of the total variance, suggesting that there is no evidence of common method bias.

## 4. Discussion

Is executive functioning (EF) related to employee well-being? This study is one of the few that sheds light on the relationship between EF and employee well-being, and the underlying mechanism. Specifically, based on several theories, a mediation model was proposed and tested to investigate the hypothetical relationships among EF deficits, work engagement, and employee well-being. All relationships are supported. 

Our analysis found a negative relationship between the ESQ-R score (i.e., EF deficits) and employee well-being. Note that the ESQ-R is a measurement of EF deficits; that is, the higher the score the poorer the EF is. Hence, the negative relationship indicates that employees who scored low in the ESQ-R (i.e., better EF) are likely to report a higher well-being score. Put differently, supporting the JD-R model [24,25], our results show that EF is helpful for employee well-being. The findings are consistent with the literature (e.g., [16,23]) that having better self-management is beneficial to one’s well-being.

Similarly, the ESQ-R score was negatively associated with the work engagement score. As explained above, the negative relationship indicates that employees who reported a better EF tend to have higher work engagement. The findings conform to the JD-R model and literature (e.g., [37,39]) that efficient resource management is helpful for employees in confronting obstacles at work and engaging with their jobs.

Alternatively, there was a positive relationship between work engagement and employee well-being. Participants who reported a higher engagement with their work also reported a higher level of well-being. The finding is not only consistent with the notion of the PERMA model [40] that engagement is essential to an individual’s well-being, but also implies that such a relationship can be generalized to the working contexts. 

The most important and unique finding of this study is the mediating effect of work engagement. Our results support that EF deficits have an indirect relationship with employee well-being via work engagement. In other words, employees with high EF deficits tend to have low engagement in work, which in turn, positively associated with well-being. To the best of our knowledge, this finding is the first empirical evidence of the hypothetical mediating role of work engagement in the relationship between EF deficits and well-being. 

The present study contributes to the literature by offering empirical evidence to the preliminary development of the EF—well-being model. The model not only reveals the benefit of EF in employee well-being but also serves as a key reference for future studies on EF in working contexts. Furthermore, we also provide insights into the underlying mechanism of the relationship between EF and well-being by demonstrating the hypothetical mediating role of work engagement. Practically, our findings suggest that EF is a promising factor to promote employees’ well-being. Management and practitioners may provide EF training or intervention such as resistance training [57] to employees to enhance their work engagement and well-being. 

Despite the promising findings, several limitations shall be emphasized. First, the present study assessed EF using a self-report which focused on five dimensions of EF (e.g., time management, emotional regulation). The results may not reflect people’s actual EF (deficits) and can be confounded by social desirability bias. Moreover, the narrow scope of the assessment overlooked other dimensions of EF (e.g., working memory). Future studies are warranted to replicate the present study using performance tasks (e.g., digit span) to assess the different dimensions of actual EF. Moreover, the replication shall involve a more diverse sample to clarify if the results continue to hold for employees in distinct industries other than academic. 

Second, this study only examined the mediating effect of work engagement. There are other factors for employees with high EF to enjoy well-being besides being engaged in work. Researchers are encouraged to expand the EF—well-being model developed in this study by exploring other possible mediators. One possible variable is rumination, the obsessive thinking characterized by excessive and recurrent thoughts that inhibit normal thinking [58]. Empirical evidence has suggested that rumination is related to executive function [59]. For example, a meta-analysis has shown that rumination is negatively related to inhibition or set-shifting [60]. On the other hand, rumination has been found to have a negative relationship with well-being [61]. Moreover, the negative relationship has also been documented in the workplace context [62]. A meta-analysis has found that an increase in rumination about workplace experience (or a lack of detachment from work) is related to poorer self-reported physical health, mental health, state well-being, and poorer task performance [63]. Taken together, it is reasonable to assume that rumination may play a mediating role in the relationship between EF and well-being among workers.

Third, the present study focused on general work engagement and overlooked the role of other specific types of engagement in workers’ well-being. For example, Tan and colleagues found that creative process engagement is positively associated with self-rated creativity [64]. The latter has had a positive association with happiness [65]. It is interesting to explore if EF of employees in creativity-related industries has an indirect relationship with their well-being via the creative process engagement and then creativity. Likewise, the relationship between work engagement and well-being deserves further attention to reveal the underlying mechanism. We refer to the literature and found that job satisfaction could be a potential mediator in the relationship. Specifically, studies have shown that work engagement has a positive relationship with job satisfaction (e.g., [66]), while the latter is positively associated with well-being (e.g., [67]). 

Finally, the cross-sectional design is limited to examining the causality of the mediation model. In the same vein, well-being can be an antecedent factor, rather than a consequence, of EF [68]. The present study is unable to clarify the possible bidirectional relationship. Future researchers are recommended to collect longitudinal data to clarify the causal relationships among EF, work engagement, and well-being. Likewise, cross-lagged panel analysis can be conducted on longitudinal data to clarify whether well-being is a predictor of EF, an effect of EF, or both. 

## 5. Conclusions

This study has two important findings: the benefit of EF for employees and a new direction for boosting employee work engagement and well-being. We provide empirical evidence that employee EF deficits have a negative and indirect relationship with their well-being via work engagement after statistically controlling the effect of self-rated creativity. Management and researchers are urged to provide EF training or intervention (e.g., resistance training; [57]) for employees. In addition, employers are also reminded to fulfill employees’ emotional, social, and physical needs, which are helpful to the development of EF [69,70]. For instance, organizations may encourage physical activities by designing the working environment to reduce sedentary behaviors or providing employee benefits to engage in physical activities. 

## Figures and Tables

**Figure 1 ijerph-19-13386-f001:**
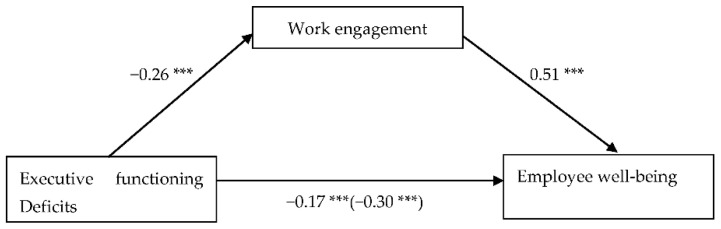
Mediation model of work engagement in the relationship between executive functioning deficits and employee well-being while controlling the effect of self-rated creativity. Note. *n* = 314 (imputed). Reported values are standardized coefficients. The total effect was shown in parentheses. The standardized indirect effect was −0.13, SE = 0.01, 95% CI [−0.15, −0.11]; *** *p* < 0.001.

**Table 1 ijerph-19-13386-t001:** Descriptive Statistics and Intercorrelations for Study Variables (*n* = 314, imputed).

Variables	Mean	*SD*	Skewness	Kurtosis	1	2	3	4
1. EFD	20.82	10.62	0.74	0.54	(0.91)			
2. WE	4.22	1.07	−0.83	0.94	−0.37	(0.93)		
3. EWB	5.39	0.98	−0.87	0.87	−0.42	0.66	(0.95)	
4. SRC	3.78	0.62	−0.45	0.55	−0.32	0.45	0.46	(0.94)

Note. *SD* = standard deviation; EFD = executive functioning deficits; WE = work engagement; EWB = employee well-being; SRC = self-rated creativity. SE skewness = 0.056; SE kurtosis = 0.113. Cronbach’s alpha coefficients were shown in parenthesis at the diagonal line. All correlation coefficients were significant at 0.001 level.

## Data Availability

The datasets generated during and/or analysis during the present study are available from the corresponding author on request.

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
