# Peer review of "The Mediating Role of Work Engagement in the Relationship between Executive Functioning Deficits and Employee Well-Being"

_ijerph, 2022, doi:10.3390/ijerph192013386_

Round 1

Reviewer 1 Report

The study is fine, and the article is well written. However, in my opinion, some revisions are needed.

  1. Diamond's model (Diamond, A. (2013). Executive functions. Annual review of psychology64, 135) should be mentioned in the introduction.
  2. The validity and reliability data of the questionnaires used should be reported. 
  3. The authors report that higher questionnaire scores correspond to higher levels of the evaluated dimension. It would be useful to report their means and standard deviations to give an idea of how the scores obtained in the various questionnaires are placed.
  4. Although the authors discuss several limitations of the study, other limitations should be discussed. First, executive functions are assessed only by a questionnaire and not by the generally used performance tasks. It would be necessary to verify whether the results are confirmed by directly evaluating the executive functions. Second, the questionnaire assesses only some executive functions (see Diamond's model). Specifying that the results exclude all executive functions not assessed would be necessary. Third, although the study evaluates well-being and the authors cited the interactions between emotions and executive functioning, this is not discussed (e.g., see the following studies: Schmeichel, B. J., & Tang, D. (2013). The relationship between individual differences in executive functioning and emotion regulation: A comprehensive review. The control within: Motivation and its regulation, 133-52. Gyurak, A., Goodkind, M. S., Kramer, J. H., Miller, B. L., & Levenson, R. W. (2012). Executive functions and the down-regulation and up-regulation of emotion. Cognition & emotion26(1), 103-118. Boncompagni, I., & Casagrande, M. (2019). Executive control of emotional conflict. Frontiers in psychology10, 359.)

Reviewer 2 Report

Dear Authors,

I have had the opportunity to review the paper titled "The Mediating Role of Work Engagement in the Relationship between Executive Functioning Deficits and Employee Well-Being". It is very well written and focuses on interesting topic of work engagement which is crucial issue for all employers nowadays. This is what, in my opinion, is missing in the beginning of the introduction. There is no general picture of the situation.

I would suggest changing the method section as I do not accept only referring to the previous research of coauthors  (Nasir et al). I would provide the regular description of the group which have been analyzed, the way it has been chosen which is know unclear and the current description raises the doubts as according to the representativeness of the sample which could be discrediting for this research and the paper. This needs to be clarified. I have also one more doubt according the sample - 66% works in the academic field which seems to be over representative if we consider the population of Malysia. Maybe the research should be dedicated to this group of employees and it should be stressed in the title and research objectives?

Best regards,

Dorota Kleszczewska

Reviewer 3 Report

The Mediating Role of Work Engagement in the Relationship between Executive Functioning Deficits and Employee Well-Being of Chee-Seng Tan and collegues is an interesting paper. However, some changes should be adressed.

Authors, on the conclusion section, should say something about the influence and role of creativity.

Authors should say more about subjects' occupation, if possible, and be more detailed about this.

The conclusion section is very poo. Authors should say more, for example, on the practical impact of the results obteined and more.

Round 2

Reviewer 2 Report

Dear Authors,

Thank you for you explanations included in the letter and the improvements made within the text.

I do not have any more remarks and I accept the paper for publication.

Kind regards,

Reviewer